# Genome-Wide SSR Markers Reveal Genetic Diversity and Establish a Core Collection for Commercial *Hypsizygus marmoreus* Germplasm

**DOI:** 10.3390/jof11120842

**Published:** 2025-11-28

**Authors:** Yan Li, Heli Zhou, Junjun Shang, Chenli Zhou, Jianing Wan, Jinxin Li, Wenyun Li, Dapeng Bao, Yingying Wu

**Affiliations:** 1National Engineering Research Center of Edible Fungi, Key Laboratory of Applied Mycological Resources and Utilization, Ministry of Agriculture, Institute of Edible Fungi, Shanghai Academy of Agricultural Sciences, Shanghai 201403, China; liyan03@saas.sh.cn (Y.L.); 13608156013@163.com (H.Z.); shangjunjun@saas.sh.cn (J.S.); zhouchenli@saas.sh.cn (C.Z.); wanjianing@163.com (J.W.);; 2Shanghai Finc Bio-Tech Inc., Shanghai 200400, China; lijinxin82@163.com

**Keywords:** *Hypsizygus marmoreus*, SSR markers, fingerprinting, phylogeny, core collections

## Abstract

Core germplasm, a strategically selected subset of the original germplasm, aims to maximize the representation of genetic diversity within the entire collection. Establishing a germplasm resource bank is essential for the effective management and sustainable utilization of genetic resources. This study developed a core germplasm repository for *Hypsizygus marmoreus*, a commercially important mushroom species, to capture the genetic diversity of the original collection with a minimal sample size. Genetic diversity and cluster analyses were conducted on 57 representative strains of *H. marmoreus*, including both cultivated and wild accessions from different regions, using 15 pairs of simple sequence repeat (SSR) markers. DNA molecular identity cards were generated for all germplasms, and cultivation trials with agronomic trait assessments were performed on 24 core accessions. A total of 115 distinct alleles were identified, with genetic similarity coefficients ranging from 0.70 to 1.00. Clustering at a similarity threshold of 0.76 classified the strains into five groups. The core germplasm panel, comprising 24 accessions (42.11% of the total collection), retained full allelic diversity and preserved the genetic and phenotypic variability of the original population, confirming its suitability for parental selection in breeding programs. unique molecular identity codes were developed for each *H. marmoreus* germplasm by integrating SSR marker profiles with data on geographical origin, fruiting body color, and cultivation traits. These were converted into DNA molecular ID codes, providing a reliable system for rapid identification and traceability of germplasm resources. The findings offer a valuable reference for breeding improvement and the protection of edible fungal varieties with independent intellectual property rights.

## 1. Introduction

*Hypsizygus marmoreus*, commonly known as beech mushroom or Bunashimeji, belongs to the phylum Basidiomycota, class Agaricomycetes, order Agaricales, family Lyophyllaceae, and genus *Hypsizygus* [1]. *H. marmoreus* is a flavorful edible and medicinal mushroom that has been industrially cultivated in Japan, China, and other Asian countries in recent years. With the rapid development of the *H. marmoreus* industry, the demand for superior varieties has grown substantially. This species exhibits significant global market demand, particularly in Asian countries such as China and Japan, which serve as both major producers and consumers. Its products are not only consumed as edible fungi but are also developed into health supplements, functional foods, and cosmetic raw materials [2]. China, as the world’s largest producer of edible fungi, maintains an annual output exceeding 45 million tons. The edible mushroom industry plays a vital role in global poverty reduction, especially through its integration into agricultural industrialization programs in developing regions. Genetic variation forms the basis of breeding, yet the limited genetic diversity of *H. marmoreus* constrains variety improvement and disease resistance development [3]. Existing *H. marmoreus* varieties still have potential for further enhancement in yield, stress tolerance, and nutritional quality. Moreover, growing consumer demand for the quality and functionality of edible mushrooms is placing increasingly stringent requirements on current varieties [4]. Therefore, a comprehensive survey, collection, and organization of existing wild and cultivated *H. marmoreus* resources in China are essential. Employing molecular marker techniques to assess the genetic diversity of different *H. marmoreus* strains will provide valuable insights for breeding programs. Genetic diversity in germplasm resources serves as the fundamental basis for variety improvement. It is a core attribute of germplasm resources and plays a decisive role in their evaluation and utilization.

The primary methods for evaluating genetic diversity include phenotypic, cytological, biochemical, and molecular marker assessments. Phenotypic and cytological approaches are cost-effective but exhibit limited accuracy, while biochemical methods are constrained by the availability of markers. In contrast, molecular markers offer high precision but require substantial costs and technical expertise. Whole-genome-based simple sequence repeat (SSR) primer design enables the direct targeting of specific genomic regions within the species of interest, significantly improving marker specificity and polymorphism through precise sequence alignment and locus screening. Nie et al. [5] conducted a genetic diversity analysis of Chinese chestnut (*Castanea mollissima*) germplasm resources using SSR markers. Based on genetic distances and relationships, they selected a representative core collection. The results demonstrated that SSR markers, a widely used molecular marker system, can effectively reveal genetic variation and phylogenetic relationships among germplasm resources. Lin et al. [6] evaluated the genetic diversity of Pleurotus spp. using molecular markers in combination with chemical composition analysis. Phylogenetic analysis revealed that the 132 tested Pleurotus samples were divided into five distinct clades. Wang et al. [7], through genome resequencing and transcriptome analyses, elucidated the genetic diversity of *Wolfiporia cocos* and identified genes associated with high yield, thereby providing an important reference for the genetic improvement and application of this species.

Germplasm resources form the foundation of breeding programs, with germplasm banks serving as repositories for genetic diversity—comparable to gene banks [8,9]. Comprehensive investigation, identification, and evaluation of collected germplasm resources are essential to maximize their utilization [10]. The concept of core germplasm, first proposed by Frankel and Brown [11], has been continually refined. Researchers worldwide have now established core germplasm collections for major economic crops such as rice, maize, soybean, and wheat [12,13,14]. These collections facilitate gene mapping and marker-assisted breeding, providing efficient pathways for further research and genetic improvement.

However, research on the establishment and application of a core germplasm collection for *H. marmoreus* has not yet been reported. In this study, a comprehensive set of *H. marmoreus* varieties was collected, and SSR molecular markers were employed to evaluate the genetic diversity and phylogenetic relationships among the assembled germplasms. Subsequently, a core germplasm set was identified, and species-specific molecular identity profiles were developed based on SSR fingerprinting, strain origins, and cultivation characteristics. In addition, cultivation trials were conducted on selected core accessions to assess their agronomic performance, aiming to provide valuable insights for hybrid parent selection and marker-assisted breeding in the development of domestically bred industrial *H. marmoreus* varieties. Establishing a core germplasm bank for *H. marmoreus* will enable the effective preservation of its genetic diversity, thereby laying a solid foundation for breeding new varieties with high yield, superior quality, and enhanced disease resistance.

## 2. Materials and Methods

### 2.1. Strains

A total of 57 *H. marmoreus* germplasms from various regions, including China, Japan, and Southeast Asia, were used in this study, as listed in Appendix A. These germplasms comprised 57 artificially strains, of which 39 were brown and 18 were white varieties. All strains were preserved in the seed bank of the Institute of Edible Fungi, Shanghai Academy of Agricultural Sciences.

### 2.2. Genomic DNA Extraction

Genomic DNA was extracted from *H. marmoreus* mycelia using the DNeasy Plant Mini Kit (Qiagen, Düsseldorf, Germany) following the manufacturer’s protocol. The quality and concentration of the extracted DNA were assessed using 1% agarose gel electrophoresis and UV spectrophotometry. A 2 μL aliquot of each DNA sample was analyzed on the gel to evaluate purity (A260/A280 ratio) and integrity, while DNA concentration was determined using NanoDrop1000 spectrophotometer (Thermo Fisher Scientific, Wilmington, NC, USA). Prior to downstream applications, DNA concentrations were adjusted to a common reference level to ensure uniformity across all samples. DNA samples with high purity (A_260_/A_280_ ≥ 1.8) were stored at −20 °C for subsequent analyses.

### 2.3. Primer Development

Reference genome data for strain HM62 were obtained from the National Center for Biotechnology Information database (BioProject number: PRJNA638014), and SSR loci were identified using the MISA tool [15] (http://pgrc.ipk-gatersleben.de/misa/, accessed on 7 July 2025). Filtering criteria were applied as follows: mononucleotide repeats ≥ 10, dinucleotide repeats ≥ 6, trinucleotide repeats ≥ 5, and tetra-, penta-, and hexanucleotide repeats ≥ 5. Primer sequences were designed in groups according to SSR locus positions, maintaining a 200 kb interval between adjacent primers for subsequent testing. Gene location data analysis and visualization were performed using MG2C [16] (http://mg2c.iask.in/mg2c_v2.1/, accessed on 28 July 2025).

Sequences containing short tandem repeats were selected for primer design using Primer Premier 6.0, resulting in the synthesis of 20 primer pairs. However, five primer pairs exhibited multiple tri-allelic peaks, which did not meet the analytical requirements of PopGene software V1.31. Therefore, the remaining 15 primer pairs were used for polymerase chain reaction (PCR) amplification of 57 *H. marmoreus* germplasm accessions. All primers were synthesized by Sangon Biotech (Shanghai) Co., Ltd. (Shanghai, China).

### 2.4. SSR Analysis

SSR analysis was performed using fifteen pairs of SSR primers to amplify genomic DNA samples through PCR. The 10 μL PCR mixture contained 1 μL of 10× PCR buffer, 0.8 μL of deoxynucleotide triphosphates (2.5 mmol/L), 0.1 μL of HSTaq DNA polymerase (5 U/μL), 0.6 μL each of forward and reverse SSR primers (5 μmol/L), 1 μL of template DNA (20–30 ng/μL), and 5.9 μL of sterile double-distilled water. PCR was performed using a thermal cycler GeneAmp9600 (Applied Biosystems, Foster City, CA, USA) under the following conditions: denaturation at 98 °C for 2 min, followed by 35 cycles at 98 °C for 10 s, 60 °C for 10 s, and 72 °C for 10 s, and a final extension at 72 °C for 5 min. PCR products were then genotyped on an ABI 3730XL Genetic Analyzer (Applied Biosystems, Foster City, CA, USA), and the sizes of the microsatellite alleles estimated using GeneScan-500 LIZ size standard (Applied Biosystems, Foster City, CA, USA) and the software GeneMapper v5.0 (Applied Biosystems).

To comprehensively evaluate the genetic diversity of *H. marmoreus*, PopGene software was used to conduct an in-depth genetic diversity analysis [17]. Several key genetic parameters were calculated, including the polymorphic information content (PIC) to assess genetic polymorphism, the number of alleles (Na) representing the total alleles per locus, the effective number of alleles (Ne) reflecting the average number of alleles contributing to genetic variation, the expected heterozygosity (He) estimating the probability that two randomly selected alleles are different, and Shannon’s information index (I) quantifying genetic diversity based on information content. Genetic distances among samples were determined using the simple matching coefficient in NTSYS-pc v2.1e, a widely adopted method for evaluating genetic similarity [18]. Cluster analysis was subsequently performed based on these genetic distances using the unweighted pair-group method with arithmetic mean (UPGMA), a hierarchical clustering algorithm that groups samples according to genetic distance, progressively constructing a dendrogram. The resulting dendrogram visually represents the genetic relationships among *H. marmoreus* germplasms. To support further breeding and conservation efforts, the M (maximization) strategy using heuristic search was carried out as implemented in PowerCore software was employed to establish a core collection from the *H. marmoreus* accessions (CapitalBio Technology Co., Ltd., Beijing, China) [19]. A core collection, which represents a subset of a larger germplasm pool, captures the maximum genetic diversity with minimal redundancy and is crucial for the efficient utilization and long-term preservation of genetic resources.

### 2.5. Agronomic Trait Measurement of H. marmoreus

A cultivation experiment was conducted to evaluate the agronomic traits of *H. marmoreus* strains. The medium for fruiting body production comprised corn cob 40–50%, sawdust 20–30%, rice bran 10–15%, wheat bran 15–20%, and corn flour 5%, with a water content of 64–66%. Approximately 600 g ± 3% of the medium was packed into 850 mL polypropylene (PP) bottles and then subjected to steam sterilization at 121˚C for 120 min. After inoculation, bottles were transferred to culture room, and culture temperature was 21–25 °C, humidity was 75%, CO_2_ concentration was 2500–4000 ppm, and the culture period was 80–85 days. Then, the upper substrate was scraped off to induce primordium formation. The bottles were then placed in a fruiting room. The suitable temperature for fruiting was 13–15 °C, with humidity at 90–95% and CO_2_ concentration at 1000–2000 ppm. The light intensity should be 50–100 Lux for the first 1–7 days, and 200–300 Lux for the subsequent 8–20 days. The shelf lights should be turned on intermittently every day to stimulate mushroom fruiting. Each strain was planted in two baskets, with 30 bottles. Phenotypic data such as cap diameter and thickness, stipe diameter and length, yield, and the number of fruiting bodies, cap center color, cap margin color, gill color, gill arrangement, stipe color, the presence or absence of flecks, and fruiting body type for all *H. marmoreus* strains were subsequently recorded.

### 2.6. Establishment of Molecular Identification Codes for H. marmoreus Germplasm

A three-component framework was employed to develop a molecular identification code system for *H. marmoreus* germplasm. This framework consisted of fingerprint codes, trait codes, and supplementary codes. The fingerprint code was generated through a sequential amplification process using the primer pairs listed in Table 1. PCR amplification was performed on genomic DNA extracted from each *H. marmoreus* strain using these primers. The resulting amplicons were separated by molecular weight through capillary electrophoresis, and their digital profiles were obtained via standardized electrophoretic pattern analysis. Based on the amplification results of 15 pairs of primers applied to 57 *H. marmoreus* germplasms, band data for each germplasm were obtained. According to the amplified band sizes of each primer pair, the presence of a band was recorded as “1” and its absence as “0”. By concatenating the band data in a consistent order of the amplification primers, a string-based DNA molecular ID represented by “0” and “1” was generated for each H. marmoreus germplasm. Simultaneously, incorporating the characteristics and source information of the germplasms (encoding rules; see Appendix A), intuitive molecular IDs were created from the corresponding strings using QR code generation technology (https://cli.im/, accessed on 25 April 2025).

Comprehensive metadata from 57 *H. marmoreus* accessions were incorporated into the supplementary coding system, including geographical origin, morphological characteristics of fruiting bodies (such as pileus and stipe coloration), and classification data from cultivation experiments. This metadata was processed through a bioinformatics pipeline that integrated phenotypic and geographical information with molecular fingerprint data. The resulting molecular identification system combines these elements to create species-specific digital profiles, enabling accurate differentiation among *H. marmoreus* germplasms. The established molecular identification codes provide a standardized framework for the efficient management, conservation, and utilization of germplasm in *H. marmoreus* breeding programs.

## 3. Results

### 3.1. Simple Sequence Repeat Markers and Genetic Structural Analysis of H. marmoreus

Using HM62 from the NCBI database as the reference genome, SSR loci were identified through MISA, and primers were designed accordingly. The screening of primers was mainly conducted through PCR amplification and capillary electrophoresis.

A total of 1943 SSR loci were evenly distributed across the *H. marmoreus* genome, with an average interlocus distance of 22.49 kb. As shown in Figure 1, of these loci, 940 were mononucleotide repeats, accounting for 48.38% of the total SSRs, and 200 were dinucleotide repeats, representing 10.29%. The remaining SSRs included trinucleotides (36.44%), tetranucleotides (1.85%), pentanucleotides (0.97%), and hexanucleotides (3.60%). Among mononucleotide repeats, the predominant motif was thymine (T) with 10 repeats, constituting 42.7% of all mononucleotides. Dinucleotide repeats displayed four motif types: AG/CT, AC/GT, AT/AT, and CG/GC. The most frequent motif was AG/CT, with 110 occurrences (55% of all dinucleotide repeats), while CG/GC was the rarest, with only one occurrence (0.5%). Trinucleotide repeats comprised ten motif types, with AGG/CCT being the most abundant (136 repeats, 19.21% of total trinucleotides), whereas AAT/ATT was the least common (4 repeats, 0.56%). Among the twelve tetranucleotide repeat motifs, AAAG/CTTT was the most prevalent (9 repeats, 25%), while AACG/CGTT, AATG/ATTC, ACTC/AGTG, ACTG/AGTC, and AGCC/CTGG were rare, each occurring once (2.78%). The most common pentanucleotide repeat was AGAGG/CCTCT with four occurrences (2.11%). For hexanucleotides, the AACCCT/AGGGTT motif represented 6.78% of the total. Overall, the most frequent repeat motifs across the genome were AGG/CCT (13.16%), ACC/GGT (12.78%), ACG/CGT (10.75%), and AG/CT (10.64%).

### 3.2. Genetic Diversity Analysis Using SSR Markers

A total of 1943 SSR markers were identified across the *H. marmoreus* genome (Figure 2). Genome sequencing data were used to design 15 SSR primer pairs that were evenly distributed throughout the genome. These primers were employed to assess genetic variation among 57 *H. marmoreus* germplasm accessions. In total, 115 allelic loci were detected using the 15 primer pairs, averaging 7.67 alleles per pair. Among them, SSR65 exhibited the lowest allelic count (3), while SSR51 showed the highest (17). Notably, only three primer pairs amplified more than 10 alleles, indicating relatively low polymorphism in the developed SSR loci. The PIC values of the 15 loci ranged from 0.132 to 0.725 (average: 0.418). Na ranged from 1.253 to 5.591 (average: 2.626), Ne ranged from 1.166 to 4.174 (average: 2.145), He ranged from 0.142 to 0.760 (average: 0.479), and Shannon’s information index (I) ranged from 0.271 to 1.524 (average: 0.824). Collectively, these results suggest that the 57 *H. marmoreus* germplasm accessions exhibit relatively high genetic diversity.

To elucidate the genetic and evolutionary relationships among the collected *H. marmoreus* germplasms, a phylogenetic tree was constructed using the UPGMA method, as shown in Figure 3A. The genetic similarity coefficients among the 57 *H. marmoreus* germplasms ranged from 0.70 to 1.00. When a similarity coefficient threshold of 0.76 was applied, all *H. marmoreus* accessions were classified into five distinct groups (Groups I–V). Group I was further divided into two subgroups, I-1 and I-2. Subgroup I-1 comprised eight brown *H. marmoreus* germplasms, including HM13, HM2, and HM24, whereas all 18 white germplasms clustered in subgroup I-2. Group II consisted of three germplasms—HM55 and HH8, derived from Chinese industrial strains, and HM47, collected from Malaysia. Group III included 13 brown germplasms, among which HZ23 and HM51 originated from Malaysia, HZ17 from Japan, and Finc-N-11 and Finc-B-6 from Chinese factories, although the latter two were originally derived from Japanese Agricultural Cooperatives. Despite their diverse geographical origins, these strains exhibited close genetic relationships. Group IV contained seven *H. marmoreus* germplasms preserved in breeding laboratories, with HM6 forming a distinct branch within this group. Additionally, the germplasm HZ18, collected from Nagano Prefecture, Japan, formed an independent branch (Group V), indicating substantial genetic divergence and a distant phylogenetic relationship compared with the other strains. Overall, SSR markers developed from the 15 primer pairs effectively distinguished the 57 *H. marmoreus* germplasms. Brown strains were primarily grouped into Groups I-1, II, III, IV, and V, while white strains were predominantly clustered in Group I-2.

Based on PowerCore analysis, a core germplasm bank comprising 24 core accessions was established from a total of 57 *H. marmoreus* germplasms. This core collection retained 42.10% of the original samples, with 68.42% derived from commercially cultivated *H. marmoreus* strains and 31.57% from artificially cultivated ones. Using phylogenetic relationships and genetic similarity coefficients, the 24 core germplasms were divided into three major clusters, represented by the colors red, blue, and green. The red cluster encompassed three germplasms conserved in the breeding laboratory, primarily corresponding to Groups I-1 and IV, as illustrated in Figure 3B. The blue cluster consisted of 11 germplasms—10 from Group I-2 and one, HM55, from Group II. Group I-2 included one strain maintained in the breeding laboratory, four supplied by Hualv Company, one from Fengke Company, three cultivated in Malaysia, and one from Japan. According to the UPGMA core germplasm map and strain provenance data, HM55 clustered with the blue group on a single branch, whereas HZ18 was associated with the green cluster. The green cluster contained 10 germplasms: seven from Group III, two from Group I-1, and one from Group V. Among these, three were sourced from Japan, one from Malaysia, five from Company A in China, and one from Company B. The clustering of germplasms originating from different regions and companies on the same branch indicates close genetic relationships among these accessions (Figure 3B).

The DNA molecular identification of *H. marmoreus* was conducted by amplifying genomic DNA using 15 primer pairs (Table 1), followed by separation of the resulting fragments through 1.5% (*w*/*v*) agarose gel electrophoresis. The fragment patterns were subsequently digitized to generate unique DNA molecular fingerprint codes, which served as identifiers for individual germplasm samples. To enhance the resolution and discrimination among different *H. marmoreus* accessions, supplementary codes were assigned to the original 57 germplasm samples. These supplementary codes incorporated specific genetic profiles, fruiting body color characteristics (pileus and stipe), and geographical origins, as detailed in Appendix A. Each *H. marmoreus* germplasm was then assigned a molecular identification (ID) consisting of its fingerprint code and supplementary code. These molecular IDs were converted into two-dimensional DNA molecular ID codes specific to each germplasm using specialized software (Figure 4). Comparison of fingerprint data revealed that four groups of germplasms shared identical molecular IDs, indicating potential limitations of the current system and highlighting the need for further refinement to ensure unique identification of all germplasm resources.

The molecular ID system developed in this study provides a standardized and efficient framework for the identification and management of *H. marmoreus* germplasm. By integrating genetic and phenotypic data, the system improves the accuracy of strain differentiation and supports germplasm preservation, breeding programs, and quality control in commercial *H. marmoreus* cultivation. Future research should aim to increase the discriminatory power of this molecular ID system through the inclusion of additional genetic markers or optimization of the encoding algorithm.

The molecular identification codes of HM2, HM12, and HM24 were found to be identical, as were those of HM5, HM8, and HM9. These two groups originated from brown-colored germplasms preserved in the breeding laboratory, suggesting possible transcription errors in strain labeling. Similarly, HM35 and HM57—sourced from Company A and Company C, respectively—shared identical DNA molecular ID code data, indicating that different companies may have assigned distinct names to genetically identical strains. The DNA molecular ID code information of HM29 and HM49 also matched; both were white *H. marmoreus* specimens collected in Malaysia. Considering that most Malaysian *H. marmoreus* cultivators import strains from China or Japan, it is likely that these two samples originated from the same parental lineage.

The identical molecular fingerprints observed among these four groups indicate extremely close genetic relationships. The consistency of Appendix A, including geographical origin and fruiting body characteristics, further supports their classification as identical or closely related varieties. Interestingly, several germplasms exhibited indistinguishable fingerprint codes, with differentiation possible only through additional metadata. Despite these highly similar genetic backgrounds, *H. marmoreus* germplasms are distributed across wide geographic regions. This widespread distribution may result from unregulated strain introductions during extended cultivation periods or from unintentional dispersal of cultivated spores into natural habitats.

Hybridization events are presumed to have occurred within *H. marmoreus* lineages, involving interbreeding between crab-flavor and white jade strains. Consequently, certain crab-flavor germplasms exhibit phenotypic variability in pileus coloration following cultivation, which may be attributed to differential expression of genes associated with pigment biosynthesis.

### 3.3. Agronomic Traits of Core Collection Cultivation

The quantitative traits evaluated included cap diameter and thickness, stipe diameter and length, yield, and the number of fruiting bodies. The qualitative traits encompassed cap center color, cap margin color, gill color, gill arrangement, stipe color, the presence or absence of flecks, and fruiting body type. Strain HM13 exhibited a prolonged fruiting cycle compared with the standard cultivation period, resulting in unsuccessful fruiting and the absence of harvestable bodies; consequently, only 23 strains were successfully harvested under normal conditions.

To demonstrate that the core collection represents the diverse germplasm resources of *H. marmoreus*, a heatmap was employed to visualize the principal agronomic traits obtained from the cultivation experiment. Figure 5 displays the quantitative traits of the 24 *H. marmoreus* strains in the core collection, including cap diameter, cap thickness, stipe diameter, stipe length, yield, and number of fruiting bodies; detailed data are provided in Appendix A.

As illustrated in Figure 5, cap diameter predominantly ranged from 10.15 to 22.35 mm, cap thickness from 4.37 to 7.61 mm, stipe length from 36.7 to 94.56 mm, stipe diameter from 5.73 to 22.97 mm, number of fruiting bodies from 24.23 to 114.52, and yield from 57.86 to 241.3 g. A yield greater than 170 g was considered a critical production threshold for factory-scale cultivation. Notably, approximately 50% of the core collection—comprising 11 strains—achieved yields exceeding 170 g, with HM55 (227.4 g), Finc-B-3 (227.8 g), and HH2 (241.3 g) surpassing 200 g. The broad variation observed across these quantitative traits indicates substantial genetic and phenotypic diversity among the core strains.

Other qualitative traits, including cap shape, cap center color, cap margin color, gill color, gill arrangement, and stipe color, are presented in Figure 6, with corresponding data available in Appendix A. Furthermore, Figure 5 and Appendix A illustrate pronounced morphological variation among the core germplasm accessions. The cap shape was predominantly round (73.91%), whereas diversity among white strains was primarily reflected in variations in the central and marginal cap colors, accounting for 26.08%. For example, Finc-B-3 and HZ22 exhibited turtle cracks, with Finc-B-3 displaying more distinct fissures compared with the other four white strains, which exhibited faint or no cracks. Among the brown strains, turtle cracks were frequently observed on the cap and categorized into three grades based on size. Strains HM6, HM30, HM55, HM31, and HZ11 exhibited large turtle cracks, while HM3, HM22, HM35, and HZ23 displayed smaller ones. The remaining eight strains also showed cap cracking, with HM37, Finc-N-11, and Finc-B-6 characterized by dense punctate patterns. The gill pleats were predominantly wavy and varied with respect to stipe length and diameter. Based on Figure 6, clear variability in pileus morphology was observed across different strains. Figure 6A provides a comparative visual analysis of the overall differences in fruiting body morphology among these strains. These observations further validate the effectiveness of SSR molecular markers in evaluating genetic diversity through the combined assessment of quantitative and morphological traits in the core germplasm of *H. marmoreus*.

## 4. Discussion

In this study, based on the genome-wide distribution analysis of SSRs in *H. marmoreus*, 20 primer pairs were initially selected to evaluate genetic variation among 57 *H*. *marmoreus* accessions. Of these, 15 primer pairs successfully amplified target loci, detecting a total of 115 alleles across all isolates, with an average of 16.38 alleles per primer pair. Genetic similarity coefficients among the 57 *H*. *marmoreus* germplasm accessions ranged from 0.70 to 1.00. When the genetic similarity coefficient threshold was set at 0.76, the *H*. *marmoreus* resources were divided into five distinct groups (Groups I–V). Thus, differentiation among the 57 accessions was effectively achieved using 15 SSR markers. PowerCore analysis further enabled the establishment of a core collection consisting of 24 accessions, representing 42.10% of the original germplasm. Notably, this core collection comprised 68.42% industrially cultivated strains and 31.57% artificially cultivated strains. He (expected heterozygosity) constitutes a pivotal indicator in genetic-diversity analyses, serving to quantify the genetic variation within populations. The normal range of He (expected heterozygosity) varies depending on species and population, typically ranging from 0.1 to 0.8. The mean genetic diversity parameters He of 57 *H. marmoreus* germplasms was 0.479, so the genetic diversity only moderate in this experiment.

Unfortunately, based on the phylogenetic analysis of 57 *H. marmoreus* based on UPGMA, strains based on geographic origin and cap color cannot cluster together, this result also demonstrated the close relationship between China and Japan; to accelerate the breeding of *H. marmoreus*, it is necessary to exchange germplasm resources globally.

Using SSR molecular marker-based genetic diversity analysis, a specific molecular ID card system was developed for *H. marmoreus* germplasm, incorporating comprehensive fingerprint data. Unlike previous fingerprinting methods used for Agaricus bisporus and Auricularia auricula, this molecular ID card system integrates SSR molecular fingerprints with supplementary coding elements inspired by human and crop genetic identification frameworks. These supplementary codes include germplasm origin, fruiting body color, and cultivation characteristics—parameters that enhance the reliability, specificity, and comprehensiveness of the molecular ID card system. For convenient identification, the alphanumeric codes were transformed into two-dimensional DNA molecular ID codes that can be rapidly scanned using mobile devices. This molecular ID system not only supports the authentication and traceability of *H. marmoreus* varieties with proprietary breeding rights but also facilitates their efficient identification, preservation, and management within germplasm repositories.

The core collection, as a representative subset of the original germplasm, serves as a fundamental basis for the evaluation and utilization of genetic resources. For such a collection to be effective, it must exhibit representativeness, genetic diversity, and specificity. A widely adopted approach to constructing a core collection involves screening genetically similar lines using genetic similarity coefficients or genetic distance matrices, followed by iterative clustering and selective elimination of redundant accessions [20]. Cosson et al. [21] identified 96 highly polymorphic SSR markers that facilitated quantitative trait locus mapping in 24 Arabidopsis core germplasms. Similarly, Nie et al. employed SSR markers to characterize 146 chestnut accessions and successfully identified 45 core collections representing the genetic diversity of chestnut germplasm. Moreover, ISSR and RAPD markers have been used to assess the genetic diversity of 86 olive germplasm resources across their primary distribution regions, leading to the establishment of a representative core germplasm set. In the present study, 57 *H. marmoreus* accessions were genetically characterized using 15 newly developed SSR markers, resulting in the establishment of a core germplasm library comprising 24 accessions. Phenotypic analysis of the *H. marmoreus* strains within this core collection demonstrated that it effectively represented the genetic diversity present in the original population with optimal efficiency. The identification of diverse germplasm resources and elucidation of phylogenetic relationships among germplasm groups provide essential insights and establish a strong foundation for future breeding initiatives. Although SSR markers exhibit high polymorphism, they are primarily sensitive to variations in non-coding regions and may show limitations in studying certain functional genes. This could result in the failure to identify some important genetic loci, thereby reducing research precision. In this study, SSR markers are useful for diversity assessment and core collection construction; however, they are insufficient to create a unique fingerprint for each strain. Developing markers with higher precision, combining transcriptome sequencing and gene functional annotation to enhance marker coverage and functional relevance, is a direction for future work. These findings hold significant implications for the global conservation and sustainable utilization of edible fungal germplasm resources.

## Figures and Tables

**Figure 1 jof-11-00842-f001:**
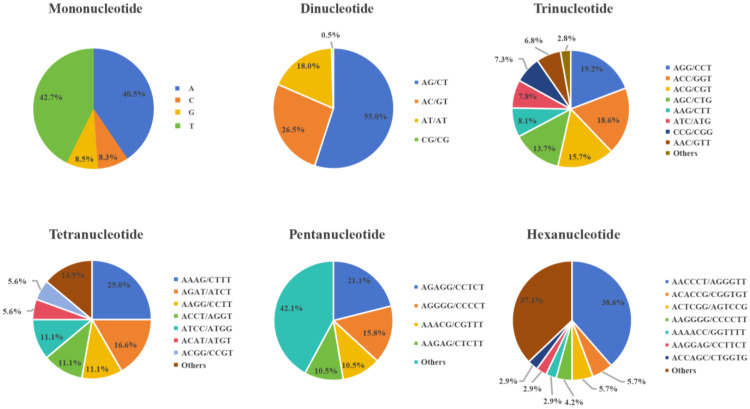
Proportions of different types of simple sequence repeat (SSR) units in the *H. marmoreus* genome. Each pie chart represents the SSR loci with different repeat units in the genome.

**Figure 2 jof-11-00842-f002:**
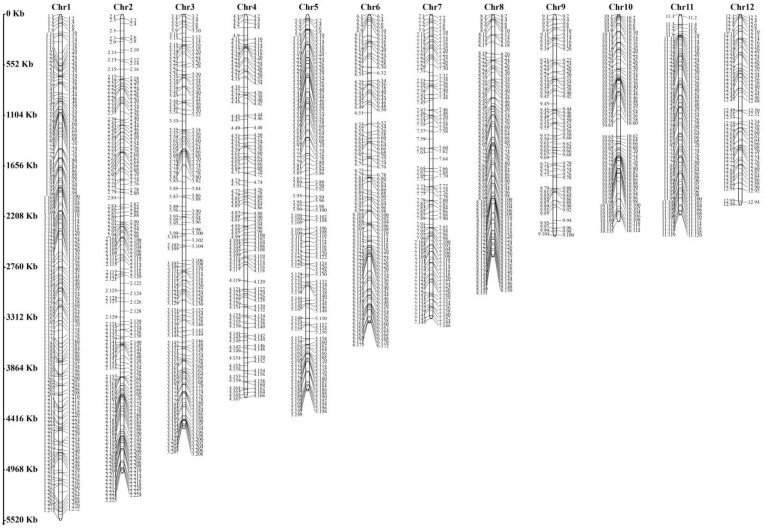
The SSR markers distribution in the chromosomal of *H. marmoreus*.

**Figure 3 jof-11-00842-f003:**
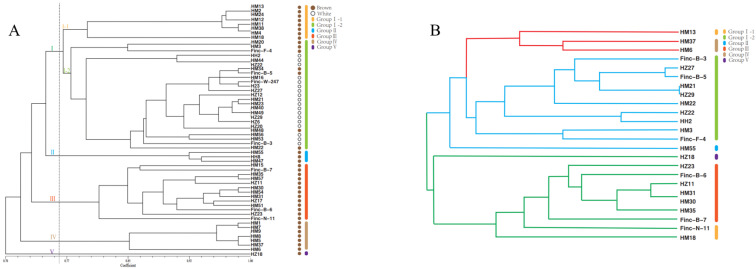
Phylogenetic analysis of 57 original germplasms (**A**) and 24 core collections (**B**) *H. marmoreus* based on UPGMA.

**Figure 4 jof-11-00842-f004:**
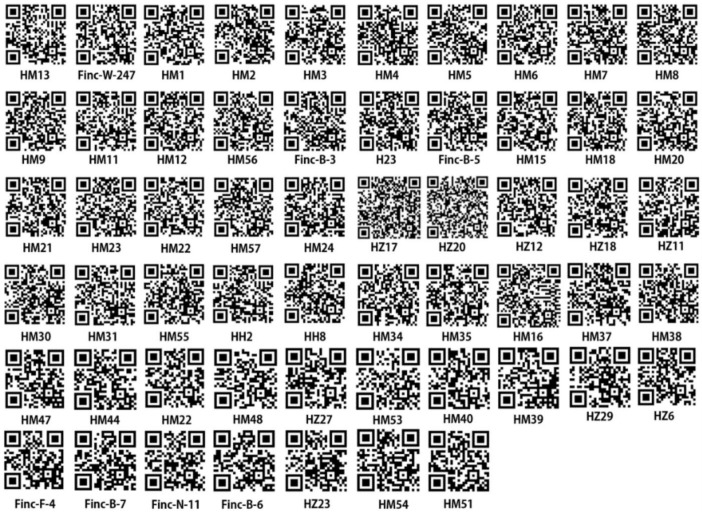
Two-dimensional codes of DNA molecular ID of 57 *H. marmoreus* (https://cli.im/, accessed on 25 April 2025).

**Figure 5 jof-11-00842-f005:**
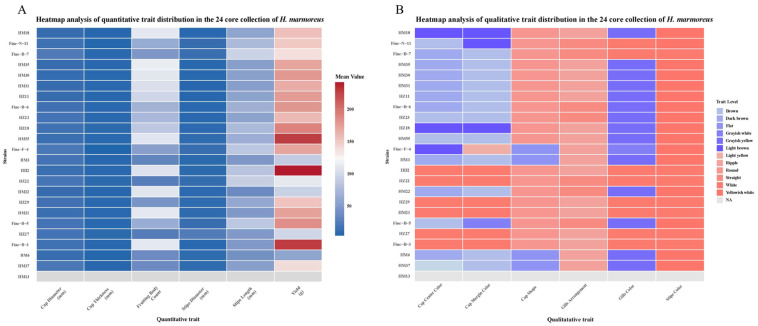
Heatmap of 24 core collections based on main quantitative traits (**A**) and qualitative traits (**B**) of *H. marmoreus*.

**Figure 6 jof-11-00842-f006:**
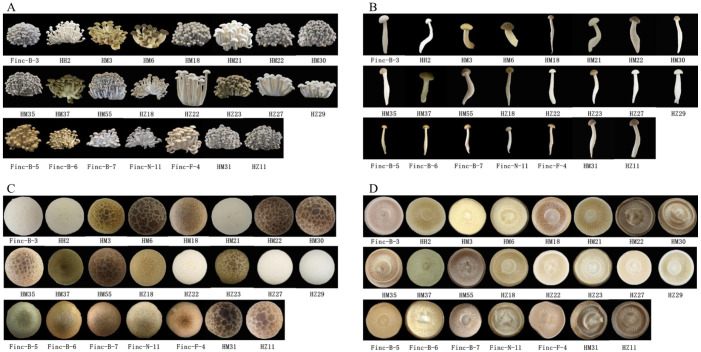
Agronomic traits of 24 core collections of *H. marmoreus* ((**A**), entire fruiting body; (**B**), single fruiting body; (**C**), the front of cap; (**D**), the back of cap).

**Table 1 jof-11-00842-t001:** Genetic diversity parameters of germplasm based on 15 pairs of SSR primers.

Locus Name	Primer Sequence (5′-3′)	PIC	Na	Ne	He	I
SSR3	F:GCGAAACCATATTCATGCGT	0.476	5	2.283	0.562	0.916
	R:ATTACTTGTCGTCGGCAGGA					
SSR11	F:GGAGTTTGAGTTGAGGCAGC	0.725	9	4.174	0.760	1.524
	R:ATGAACCAGACCAAAGACCG					
SSR12	F:GGCACGGACATAGACCTCAT	0.325	5	1.690	0.408	0.598
	R:GTGGTGGTGTGACGACGTAT					
SSR15	F:GATTGTTCGCTGGAACACCT	0.436	6	2.109	0.526	0.846
	R:ACTCACGATGAAGGCAAACG					
SSR18	F:GAGGATTGAAGGGACTGTCG	0.132	7	1.166	0.142	0.271
	R:CCTCATCCTCCGACTCTACG					
SSR22	F:TGCTGGTGAGTGAGTTGGAG	0.374	4	1.988	0.497	0.690
	R:ACGGCGACATAATCTGCTCT					
SSR26	F:GTGATTGGGTTCGTGTCGTC	0.339	9	1.563	0.360	0.734
	R:ACCTCGAGCTCAACTTCTGC					
SSR32	F:AACCTCCAGTCACAACCTGC	0.171	8	1.232	0.188	0.336
	R:CCTTGCTTCTTGTCGGATGT					
SSR35	F:TCCGTGAGAGGACGGAGTAG	0.592	11	2.840	0.648	1.185
	R:CTCAGCAACGACGAACAACC					
SSR36	F:TCTTCTTGTAGAGCGCCTCG	0.586	10	2.658	0.624	1.241
	R:CTCTCGACGCGTGTTCCT					
SSR38	F:TCTTCTTGTTCGGCGGTATC	0.323	7	1.590	0.371	0.639
	R:ATCCGGCACAGGTAAAGATG					
SSR48	F:GCCGTGTGACACCAAATACA	0.523	9	2.513	0.602	0.992
	R:ACATCTAACCGACCCACGAC					
SSR51	F:CACCCCCTCATCCTACACAT	0.582	17	2.909	0.656	1.082
	R:TGGTCAGAAGAAACGTCGTG					
SSR61	F:CTTCCTTGTCCCGACATGAT	0.360	5	1.757	0.431	0.701
	R:GGCTCGGCCTTAGTGTGTAA					
SSR65	F:CTGCTCCTCGTTCTTTGAGC	0.327	3	1.702	0.413	0.603
	R:GGTAAGGTCTGGGACGGATT					
Total		6.271	115	32.174	7.189	12.360
Mean		0.418	7.67	2.145	0.479	0.824

PIC, polymorphic information content; Na, the number of alleles; Ne, the effective number of alleles; He, the number of expected heterozygosity; I, Shannon’s information index.

## Data Availability

The original contributions presented in this study are included in the article and Appendix A. Further inquiries can be directed to the corresponding authors.

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
