# Peer review of "Genome-Wide SSR Markers Reveal Genetic Diversity and Establish a Core Collection for Commercial Hypsizygus marmoreus Germplasm"

_jof, 2025, doi:10.3390/jof11120842_

Round 1

Reviewer 1 Report

Comments

  1. The two-stage PCR program (35 cycles at 60°C annealing, then 10 cycles at 53°C) is highly unusual and not standard practice for SSR amplification. The authors must provide a robust scientific justification for this protocol. What was the purpose of the second, lower-stringency cycling? Without justification, it raises concerns about the specificity of the amplification and the quality of the resulting fragment data.
  2. The method states PCR products were analyzed by capillary electrophoresis but omits critical details. What was the instrument? What was the internal size standard used? How were alleles binned? Mentioning GeneMapper software is good, but the specific parameters and quality controls for allele calling are missing.
  3. While the use of PowerCore is noted, the parameters used for its execution are not specified. What proportion of the total collection was the core set intended to represent? What was the primary criterion for selection? This needs to be detailed.
  4. The distinction between "39 industrially cultivated strains" and "18 artificially cultivated strains" is confusing and likely a mistranslation or error. In standard mycological context, all cultivated strains are "artificial." The intended distinction is probably between commercial cultivars and wild-collected isolates that have been cultured. This must be clarified, as the genetic diversity between wild and cultivated populations is a key point of interest.
  5. The cultivation experiment is described, but it lacks essential details for a scientific study.
  6. The substrate recipe is a complex mixture, but no specific proportions are given. Was the experiment replicated? If so, how many biological replicates per strain? The environmental conditions (light, humidity) in the fruiting chamber are critical for mushroom development and should be specified. The specific "phenotypic data" recorded must be listed.
  7. The process for converting electrophoretic patterns into a "unique digital representation" or "fingerprint code" is not explained. Is it a simple binary matrix (1/0 for presence/absence of bands), or a more complex code based on allele sizes? A clear, standardized algorithm for generating this code is necessary for it to be useful to other researchers.
  8. The study provides a full set of genetic diversity parameters (PIC, Na, Ne, He, I) for all 15 SSR loci, which is excellent.
  9. Section 3.1 begins with a detailed description of genome resequencing, alignment, and variant calling (lines 214-229). This entire paragraph belongs in the Methods section. Its presence here is confusing and obscures the actual results. The only result in this paragraph is the mention of "five distinct subgroups," which is then not mentioned again until the phylogenetic tree in Figure 3. This information should be integrated with the UPGMA results in 3.2.
  10. The discussion simply restates the results (e.g., "15 primer pairs...detecting a total of 172 alleles") without providing a deeper interpretation. There is a critical discrepancy here: In the Results (Table 1), the total number of alleles was 115, not 172. This must be corrected. Beyond this error, the discussion should answer "why?" Why was the genetic diversity only moderate? Does the clustering into five groups reflect geographic origin, cultivation history, or color phenotype? The close genetic relationship between Japanese and Chinese factory strains (mentioned in Results) is a fascinating finding that deserves discussion in the context of global strain exchange and breeding programs.
  11. The Discussion completely ignores the most significant limitations revealed in the Results. Most critically, it does not mention that the molecular ID system failed to uniquely distinguish all accessions. A robust discussion must acknowledge that the 15 SSR markers, while useful for diversity assessment and core collection establishment, were insufficient for creating a unique fingerprint for every strain. This should be framed as a direction for future work

Comments

  1. The two-stage PCR program (35 cycles at 60°C annealing, then 10 cycles at 53°C) is highly unusual and not standard practice for SSR amplification. The authors must provide a robust scientific justification for this protocol. What was the purpose of the second, lower-stringency cycling? Without justification, it raises concerns about the specificity of the amplification and the quality of the resulting fragment data.
  2. The method states PCR products were analyzed by capillary electrophoresis but omits critical details. What was the instrument? What was the internal size standard used? How were alleles binned? Mentioning GeneMapper software is good, but the specific parameters and quality controls for allele calling are missing.
  3. While the use of PowerCore is noted, the parameters used for its execution are not specified. What proportion of the total collection was the core set intended to represent? What was the primary criterion for selection? This needs to be detailed.
  4. The distinction between "39 industrially cultivated strains" and "18 artificially cultivated strains" is confusing and likely a mistranslation or error. In standard mycological context, all cultivated strains are "artificial." The intended distinction is probably between commercial cultivars and wild-collected isolates that have been cultured. This must be clarified, as the genetic diversity between wild and cultivated populations is a key point of interest.
  5. The cultivation experiment is described, but it lacks essential details for a scientific study.
  6. The substrate recipe is a complex mixture, but no specific proportions are given. Was the experiment replicated? If so, how many biological replicates per strain? The environmental conditions (light, humidity) in the fruiting chamber are critical for mushroom development and should be specified. The specific "phenotypic data" recorded must be listed.
  7. The process for converting electrophoretic patterns into a "unique digital representation" or "fingerprint code" is not explained. Is it a simple binary matrix (1/0 for presence/absence of bands), or a more complex code based on allele sizes? A clear, standardized algorithm for generating this code is necessary for it to be useful to other researchers.
  8. The study provides a full set of genetic diversity parameters (PIC, Na, Ne, He, I) for all 15 SSR loci, which is excellent.
  9. Section 3.1 begins with a detailed description of genome resequencing, alignment, and variant calling (lines 214-229). This entire paragraph belongs in the Methods section. Its presence here is confusing and obscures the actual results. The only result in this paragraph is the mention of "five distinct subgroups," which is then not mentioned again until the phylogenetic tree in Figure 3. This information should be integrated with the UPGMA results in 3.2.
  10. The discussion simply restates the results (e.g., "15 primer pairs...detecting a total of 172 alleles") without providing a deeper interpretation. There is a critical discrepancy here: In the Results (Table 1), the total number of alleles was 115, not 172. This must be corrected. Beyond this error, the discussion should answer "why?" Why was the genetic diversity only moderate? Does the clustering into five groups reflect geographic origin, cultivation history, or color phenotype? The close genetic relationship between Japanese and Chinese factory strains (mentioned in Results) is a fascinating finding that deserves discussion in the context of global strain exchange and breeding programs.
  11. The Discussion completely ignores the most significant limitations revealed in the Results. Most critically, it does not mention that the molecular ID system failed to uniquely distinguish all accessions. A robust discussion must acknowledge that the 15 SSR markers, while useful for diversity assessment and core collection establishment, were insufficient for creating a unique fingerprint for every strain. This should be framed as a direction for future work

Reviewer 2 Report

The aim of this study was to develop a core germplasm repository for the mushroom Hypsizygus marmoreus to capture the genetic diversity of the original collection with a minimal sample size. The germplasm core for H. marmoreus has not yet been published. In addition, a specific molecular identification card system has been developed for the germplasm of H. marmoreus.

The purpose of the Hypsizygus marmoreus germplasm core is to create a small, representative subset of a large genetic collection that encompasses the full genetic diversity of the original population with minimal redundancy. This makes the management and use of H. marmoreus germplasm resources more efficient, accelerating processes such as mushroom breeding by providing a manageable reference set for researchers to identify and use valuable genes for traits such as yield or disease resistance, or beneficial biological activities.

The topic of this study is original. The results of this research provide a good basis that can be used in future research to establish new strains with specific biological characteristics, such as higher yield, higher nutritional content, or specific biological activity and applications in the functional food and nutraceutical industries.

This mushroom is known for its potential in the functional food industry. The genetic diversity within the germplasm core could serve as a source from which future new strains of H. marmoreus can be developed. This development may involve selection or crossing for specific germplasm traits, including increased yield and nutritional value, as well as beneficial biological activities.

The germplasm core for H. marmoreus has not yet been published. This is the first study of this type for H. marmoreus. Also in this research, a molecular identification card system was developed for the H. marmoreus germplasm, which includes comprehensive fingerprint data. In addition, alphanumeric codes have been transformed into two-dimensional DNA molecular identification codes that can be quickly scanned using mobile devices. This identification system supports the authentication and traceability of H. marmoreus strains with breeding rights, and also enables efficient identification, conservation and management within germplasm repositories. 

The methodology used in the research is adequate and there are no additional suggestions for its improvement.

The discussion and conclusion adequately interpret the significance of the findings. The conclusion briefly states the main findings of the study. 

The references are appropriate.

I have no additional comments on the tables and figures, other than those I have already attached.

There are only two minor suggestions regarding the manuscript:

  1. Should qualitative and quantitative features be labeled with the letters A and B in the legend of Figure 5? These letters are present in Figure 5.
  2. Same comment for the legend of Figure 6. Please explain Figures A-D (e.g. Figure 6A - morphology of the fruiting body).

Round 2

Reviewer 1 Report

The authors addressed all my comments. I am satisfied with the revised manuscript. I accept it in its current form

The authors addressed all my comments. I am satisfied with the revised manuscript. I accept it in its current form